# ADAPT: Actively Discovering and Adapting to Preferences for any Task

**Maithili Patel**[1], **Xavier Puig**[2], **Ruta Desai**[2], **Roozbeh Mottaghi**[2], **Sonia Chernova**[1], **Joanne Truong**[2]*, **Akshara Rai**[2]*
[1]Georgia Institute of Technology
[2]FAIR Labs, Meta
maithili@gatech.edu

## Abstract

Assistive agents should be able to perform under-specified long-horizon tasks while respecting user preferences. We introduce Actively Discovering and Adapting to Preferences for any Task (ADAPT) – a benchmark designed to evaluate agents' ability to adhere to user preferences across various household tasks through active questioning. Next, we propose Reflection-DPO, a novel training approach for adapting large language models (LLMs) to the task of active questioning. Reflection-DPO finetunes a 'student' LLM to follow the actions of a privileged 'teacher' LLM, and optionally ask a question to gather necessary information to better predict the teacher action. We find that prior approaches that use state-of-the-art LLMs fail to sufficiently follow user preferences in ADAPT due to insufficient questioning and poor adherence to elicited preferences. In contrast, Reflection-DPO achieves a higher rate of satisfying user preferences, outperforming a zero-shot chain-of-thought baseline by 6.1% on unseen users.

## 1 Introduction

Consider an embodied agent that assists users at tasks, such as "organize incoming stock" in a warehouse, or "prepare an omelet" at home. These tasks are high-level, under-specified and lack information about user preferences. For example, a warehouse user might prefer to store larger boxes towards the back or to keep frequently used items at the front for easy access, and the user in a home setting might want an omelet made in olive oil instead of butter. In contrast, grounded planning works typically assume detailed instructions (Yenamandra et al., 2023; Liang et al., 2023; Huang et al., 2023a), such as "move the small box to the first shelf". It is tedious for a user to break down high-level tasks into unambiguous step-by-step instructions, especially for multi-step, long-horizon tasks. Instead, we argue that the agent should be able to autonomously plan towards the high-level task, and actively ask questions to uncover user's preferences that are not explicitly stated in the goal command.

Actively asking questions towards an open-world long-horizon task is challenging, because it requires creating a plan in a large search space, understanding possible variations of that plan, and determining when and what to ask the user to execute their preferred variation. Recent works have showcased the abilities of large language models (LLMs) in open world planning (Chang et al., 2025; Rana et al., 2023; Huang et al., 2023b; Liu et al., 2023; Brohan et al., 2023). However, LLMs have been shown to assume, rather than ask for user preferences (Shaikh et al., 2023). Works that learn user preferences for embodied tasks (Wang et al., 2024; Wu et al., 2023; Kapelyukh & Johns, 2022; Jain et al., 2023) typically rely on task demonstrations. However, providing multiple demonstrations for all everyday tasks, such as preparing breakfast, is difficult and not scalable. Ideally, an assistive agent should be able to execute tasks autonomously, and ask questions when needed.

---

*Indicates equal contribution

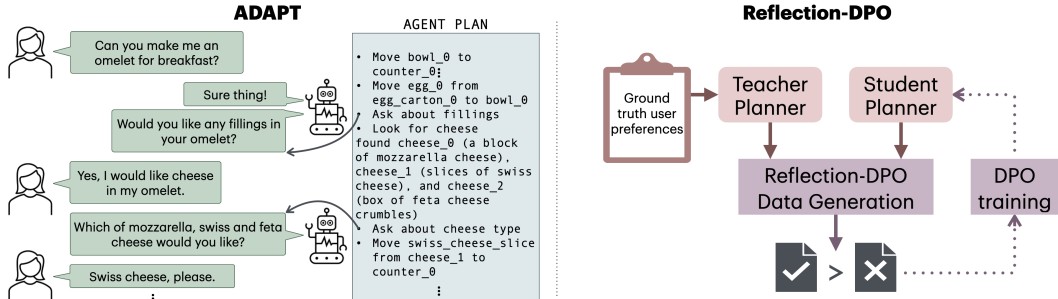

Figure 1: ADAPT (left): A new benchmark that requires an agent to actively elicit user preferences through questions. Reflection-DPO (right): A novel approach for training an LLM for active questioning using a privileged teacher, and a reflection mechanism that introduces questions into the dataset.

We contribute Reflection-DPO (Figure 1, right), an approach for teaching LLMs to intelligently ask questions regarding user preferences, where needed. Our **key insight** is that LLMs are not good at actively asking questions to uncover hidden preferences, but given access to preferences, they can adapt task execution to align with them. Reflection-DPO uses a privileged LLM (teacher), primed with knowledge about preferences of a particular user, to train a student LLM to adhere to preferences without privileged knowledge. However, as we show in Section 6.3, directly trying to imitate the teacher actions isn't enough, because the student might lack key information needed to predict the teacher's actions. To account for missing information, we introduce a reflection step that generates a candidate question to help the student predict the teacher's action. Ultimately, this teaches the student to adhere to preferences (as demonstrated by the teacher) by learning to acquire the necessary information through active questioning (by reflection), enabling it to fulfill ambiguous goals, while adhering to user preferences by proactively asking questions.

Our second contribution is a benchmark – Actively Discovering and Adapting to Preferences for any Task (ADAPT) – designed to evaluate closed loop performance of assistive agents on long-horizon, ambiguous everyday tasks (Figure 1, left). ADAPT contains a set of 8 high-level tasks and 16 personas. We instantiate a total of 128 persona-task combinations, each characterized by up to 14 preferences and grounded in scenes with on average 162 movable objects, in a text-based planning domain. We focus on common daily activities (e.g., making eggs or cereal) and support text-based actions, which can be state-changing actions or open-ended questions, such as 'Open cabinet' or 'How would you like your eggs cooked?'. An LLM is prompted to be a 'human' user that answers the agent's questions. This results in an open-ended communication benchmark with challenging, long-horizon, ambiguous tasks, useful for benchmarking adaptive assistive agents.

Our results show that SoTA LLMs such as Llama3.1-70b, when used to control agents, do not adhere to preferences in ADAPT. Instead, training using Reflection-DPO enables an LLM to ask informative questions and adapt to preferences, improving upon an untrained LLM by achieving 8.6% higher rate of preference satisfaction for unseen users. On the same metric, Reflection-DPO outperforms a zero-shot chain-of-thought LLM, our leading baseline, by 6.1%. Our key contributions[1] are: (1) A novel approach, Reflection-DPO, for training LLMs to actively ask questions and adapt task execution to align with elicited preferences. (2) A benchmark, ADAPT, for evaluating assistive agents on active questions that operates within a grounded text planning domain, and includes a large action space and user models capable of answering open-ended questions.

## 2 Related Work

**Interactive learning for behavior adaptation** has been widely studied. Interactive feedback includes action corrections (Li et al., 2025; Korkmaz & Bıyık, 2025; Pérez-Dattari et al., 2019; Liu et al., 2022), preference feedback (Yang et al., 2024; Sadigh et al., 2017; Christiano et al.,

---

[1]Code for ADAPT and Reflection-DPO is openly available at `https://github.com/Maithili/Adapt`.

2017; Bıyık et al., 2022), and language-based corrections (Liang et al., 2024; Han et al., 2025; Sharma et al., 2022; Campos & Shern, 2022), using feedback on agent-generated trajectories to understand user preferences. Some methods personalize by predicting feedback (Liang et al., 2024), and co-embedding corrective language feedback with trajectory (Yang et al., 2024). These approaches focus on single actions and struggle with long-horizon tasks due to context-length limits and learning shared embedding for multi-step trajectories. Prior work Bärmann et al. (2024) has used memory to capture online feedback, but relies explicit preferences. LLM-Personalize (Han et al., 2025) iteratively improves at multi-step object rearrangement by learning from user ratings, but relies on demonstrations and dense feedback. Natural language has also been used to express preferences (Wu et al., 2023; Wang et al., 2024), but these works also rely on prior task demonstrations. Instead of demonstrations or dense user feedback, we propose adapting through active questioning during task execution to elicit preferences.

**Task-oriented dialog** is crucial for instruction, guidance, collaboration, and clarification. Prior works have explored dialog for collaborative tasks (Bara et al., 2021; Narayan-Chen et al., 2019), paired commander-follower agents (Padmakumar et al., 2022; Thomason et al., 2019; Banerjee et al., 2020), and agent asking for help (Singh et al., 2022; Zhang et al., 2023). When the task is not explicitly defined, some works optimize directly for user attention (Yoshino & Kawahara, 2015) or enable responses to their implicit requests (Tanaka et al., 2024). Our approach differs by focusing on exchanging information about hidden user preferences, not task instructions or execution help. Follow-up work (Gella et al., 2022) that analyzes a commander-follower setup (Padmakumar et al., 2022) shows that only 1.6% of the utterances asked for additional information, and 2.7% more provided information relevant to learning hidden preferences. Methods that seek to efficiently ask for help either separately train a model through RL (Singh et al., 2022), directly eliciting an expert action rather than obtaining information, or use a measure of entropy lower-bound of the goal (Zhang et al., 2023), requiring an explicit estimate of goal uncertainty, which is difficult to obtain in open-set preferences as in our setup. Prior work (Hong et al., 2023) has utilized offline RL from experts, focusing on dialogue efficiency rather than user preferences. Instead, we gather information about hidden user preferences through dialogue.

**Information-seeking dialog** has been employed to clarify user preferences (Wang et al., 2024), disambiguate instructions (Park et al., 2023), and improve predictions when uncertain (Ren et al., 2023). Prior work on active clarifications for personalization (Wang et al., 2024) rely on preference hypotheses generated from demonstrations. Research on asking for clarification between multiple probable LLM predictions (Ren et al., 2023) is not geared to long horizon tasks, and struggles to distinguish between action-related and preference-related ambiguities. In contrast, we use a privileged teacher to teach a student LLM how to follow preferences. Most related to our work is STaR-GATE (Andukuri et al., 2024), which enables a virtual agent to act based on hidden preferences. However, this agent can only take one action based on questions, whereas our tasks involves sequential decision making. Prior benchmarks that study preference elicitation through dialog operate on domains such as online shopping assistance, evaluated on datasets of shopping prompts and target items (Andukuri et al., 2024; Piriyakulkij et al., 2023), game-like decision-making (Lin et al., 2024) or price negotiation (Verma et al., 2022), evaluated on human-human dialog datasets, and content recommendation, evaluated through user studies (Handa et al., 2024). However these domains are either purely conversational or low-dimensional game-like environments. They do not support complex tasks, such as those expected of assistive household agents, requiring grounded planning, diverse states and actions and long-term interactions.

## 3  Problem definition

We consider an agent controlled by a policy $\pi$, tasked with achieving a goal $g$ while adhering to a user's preferences $\Gamma$. At each step $k$, the agent takes an action $a_k$, given the task goal $g$, and the history of agent interactions, including previous actions $a_{0:k-1}$, observations $o_{0:k-1}$, and user responses $u_{0:k-1}$. The actions available to the agent include physical actions, questions for the user, and a termination action. Physical actions involve object manipulation

and exploration, and result in an updated state of the environment and an observation. User responses to questions provide feedback, and help uncover user preferences $\Gamma$.

Each episode results in a trajectory $\langle g, a_{0:K}, o_{0:K}, u_{0:K} \rangle$ of $K$ steps. A reward $r \in \{0, 1\}$ is assigned to the trajectory depending on how well the agent's actions satisfy the user preferences: $g, \Gamma, a_{0:K}, o_{0:K} \rightarrow r$. User preferences are task-related constraints, not captured in the goal $g$, and a priori unknown to the agent, such as preference for dairy-free milk. The policy $\pi$ aims to maximize the reward $r$. At each step $k$, $\pi$ predicts the next action $a_k = \pi(g, a_{0:k-1}, o_{0:k-1}, u_{0:k-1})$, which is executed in the environment. If the policy generates a question, the agent receives a user response. A termination action ends the trajectory.

## 4 The ADAPT benchmark

We develop agents capable of adhering to user preferences for diverse, long-horizon household tasks, such as "Preparing eggs for breakfast". We focus on meal preparation tasks, since they involve complex preferences beyond object locations. Setting up such tasks requires an environment with: 1) a grounded simulation supporting diverse actions, 2) large environmental diversity to accommodate realistic preferences, and 3) diverse user models that are characterized by preferences and can interact with the agent. We present a benchmark ADAPT that exhibits these characteristics.

**Grounded text-based simulation**: To support complex tasks, such as "make an omelet for breakfast" we develop a grounded interactive text-based simulation environment. This simulator allows for an open set of state changes, enabling complex actions such as mixing eggs and chopped vegetables to get omelet batter. The environment is represented using a textual scene graph, similar to prior work (Puig et al., 2018). Each node in the graph corresponds to an entity, such as movable objects, furniture, or room and is characterized by: 1) a unique identifier (e.g., milk_1), 2) a freeform text description (e.g., carton of oat milk), 3) state description as a list of state tags and contents (e.g. open, contains oat_milk), and 4) an optional type, designating whether the object is edible or if it is a container which can hold food. Each interaction begins with the agent receiving an scene layout, consisting of the rooms and furniture, but no objects. The agent must take text-based actions, to discover the moveable objects and their locations. Each action results in an observation, provided to the agent as feedback. For exploration actions, the observation includes a list of observed objects. For manipulation actions, the observation includes a summary of changes made to the environment, if the action was successful, else the reason for action failure. Failures can occur due to impossible actions such as heating something that is not on a cooking appliance, mixing non-food objects into a mixture etc.

**Supporting an open set of actions**: Our simulator supports an open set of actions necessary for simulating a wide range of everyday tasks, alongside a base set of actions. For open-set actions, we create two templates: "`<action> items in <container> to get <object>`" and "`<action> the <object> to get <object>`." These templates allow for actions like "mix items in bowl_0 to get omelet_batter" or "crack the egg_0 to get cracked_egg_0.". We rely on the agent to provide semantically meaningful open set actions (e.g."cracked_egg" from egg_0), and do not explicitly check the validity of each action. Using LLMs as the policy to control agents ensures that open set actions are diverse yet semantically meaningful. Given an action, the simulator transforms the nodes involved in the action into the expected output node, as specified in the template. For example, the bowl_0 node now contains omelet_batter. We also provide a base set of actions that enable state-changes, exploration and relocation, including open, close, turn on, turn off, move, move from, pour, search and look for. Successful execution of actions require certain pre-conditions to hold. All actions require the referred entities to exist in the environment. *move from* and *pour* require the referred object being in the mentioned container. Certain freeform actions require the referred object to be in the required location as a pre-condition for successful execution. See Appendix C.1 for a full list of actions and their pre-conditions.

**Task, object and scene diversity**: To support diverse user preferences, we offer a wide variety of objects, including several variations within categories, summing up to 232 movable objects, over 6 rooms and 37 static appliances and furniture. Variations within each category,

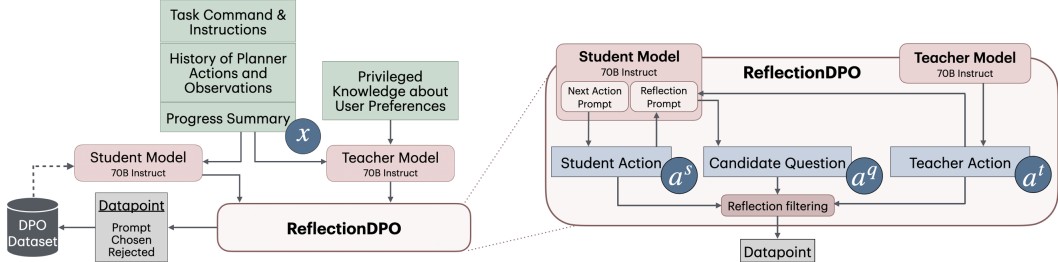

Figure 2: Training mechanism of Reflection-DPO, using a teacher model and probability-based reflection to create a dataset of desirable actions for finetuning the student using a DPO trainer

such as tea, are identified through object descriptions, such as *box of green tea bags*, *box of chamomile tea*, etc. We randomly generate each scene by including all rooms, furniture and a set of mandatory objects('napkins_0', 'salt_shaker_0', 'pepper_shaker_0', 'water_bottle_0', 'ice_tray_0'), and adding each of the other moveable objects with a 70% probability. We include 8 high-level tasks, related to breakfast preparation, such as "Make toast and coffee for breakfast" and "Prepare omelet for breakfast" (full list in Appendix C.2). Due to the large combinatorial space of scenes, generated by sampling each object, each scene used in train and test interactions is unique, though composed of the same set of objects and tasks. The high-level nature of tasks, coupled with the presence of a broad range of objects, allows several variations in each task, supporting diverse preferences, such as adding a specific category of ingredient, and preferring one ingredient variation over another.

**Modeling user personas**: Finally, we create persona models to simulate users, powered by LLMs and characterized by a set of preferences. ADAPT includes 16 personas, each with an average of 14.25 preferences, 5.5 of which apply on average to a given task (see Appendix C.3 for a complete list). Preferences include: 1) preferring certain food variations, such as non-dairy milk, 2) adding sides or toppings, such as herbs to eggs or nuts to yogurt, 3) excluding add-ons, such as no cream or sugar with coffee, 4) temporal ordering preferences, such as pouring cereal before milk, or preparing beverages before food, 5) preferred object usage, such as preferring pour-over for coffee, 6) preferred locations for serving meals, 7) preferring certain preparation alternatives, such as preferring tea iced over hot.

Each preference includes a textual description, and a verification function. We prompt an LLM with the user preferences to enable it to answer questions during task execution. The verification functions are python functions that can evaluate whether the linked preference is satisfied, violated or inapplicable in a given interaction. The applicability of a preference can be conditioned on the task at hand, such as 'when making toast use whole grain bread', and on the availability of objects, such as 'use the espresso machine, if available, else pour-over'. We track entities through an interaction to verify preference satisfaction, keeping track of object usage, additions during preparation, temporal order of usage, and final placement. Both preferences and verification functions are manually created.

## 5 Learning to plan with user preferences

To enable an LLM to learn to actively elicit user preferences while performing a task, we introduce Reflection-DPO, a training mechanism which leverages a privileged teacher model with access to user preferences. Reflection-DPO finetunes an LLM using a corrective dataset of desirable predictions, which can be physical actions or questions (Figure 2: left). To create the corrective dataset, we choose between physical actions demonstrated by the privileged teacher model, and questions demonstrated by a reflection mechanism at every timestep (Figure 2: right). At its core, Reflection-DPO has three main components: 1) Dagger-style (Ross et al., 2011) demonstration of physical actions by a privileged teacher model, 2) demonstration of informative questions by a novel reflection mechanism, and 3) Direct Preference Optimization (DPO) (Rafailov et al., 2024) for finetuning the student LLM.

**Generating data with a privileged teacher:** We use an LLM with privileged information about user preferences to create a teacher $\pi^t$ that generates data to train a student LLM $\pi^s$. First, given a set of personas and tasks, we generate interaction trajectories $\langle g, a^s_{0:K}, o_{0:K}, u_{0:K}, r \rangle$ using the student model. Next, we use the teacher model, which has access to privileged preferences $\Gamma$, to predict an action at each time-step $k$ in the student trajectory: $a^t_k = \pi^t(g, \Gamma, a^s_{0:k-1}, o_{0:k-1}, u_{0:k-1})$. Similar to Dagger (Ross et al., 2011), this results in a corrective dataset, where teacher actions $a^t_k$ provide corrections to the student actions $a^s_k$. We skip datapoints where teacher and student predict the same action.

**Reflection to generate questions:** The student model cannot directly imitate the teacher, because the teacher model has access to privileged information regarding user preferences, which the student lacks. To teach the student model to actively elicit missing information in an efficient manner, we introduce a novel reflection mechanism, outlined in Algorithm 1. The reflection mechanism queries the student model with a reflection prompt that includes the student and teacher predictions, along with the interaction history. The reflection prompt asks the student model to generate a candidate question $a^q$ that might help the student predict the teacher's action. For example, if teacher action adds almonds to the user's cereal, a candidate question might ask about desired toppings.

However, when prompted in this way, the student model tends to produce a question even if there is no missing information, and often the predicted question might not be relevant to the user preference. To ensure that the questions are useful, we compute the utility of a question in predicting the teacher action. Given interaction history $x$, we calculate question utility $\Delta_q$ as the increase in probability of predicting teacher action under the student distribution given the question: $\Delta_q = P_{\pi^s}(a^t|q, x) - P_{\pi^s}(a^t|x)$, where $P_{\pi^s}(a|x)$ is the probability of predicting an action $a$ by the student LLM given context $x$, calculated as average over tokenwise probabilities. If the question

---

**Algorithm 1** Reflection mechanism

**function** REFLECTION($P_{\pi^s}(\cdot), a^s, a^t$)
  $a^q \leftarrow \text{getQuestion}(P_{\pi^s}, a^s, a^t)$
  $\Delta_q \leftarrow P_{\pi^s}(a^t|q) - P_{\pi^s}(a^t)$
  $\Delta_t \leftarrow P_{\pi^s}(a^s) - P_{\pi^s}(a^t)$
  **if** $\Delta_t < 0$ **then**: $a_{\text{chosen}} = a^t$
  **else if** $\Delta_q > \epsilon_1$ **then**: $a_{\text{chosen}} = a^q$
  **else if** $\Delta_t < \epsilon_2$ **then**: $a_{\text{chosen}} = a^t$
  **else**: $a_{\text{chosen}} = \text{None}$
  **end if**
  **return** $a_{\text{chosen}}$
**end function**

---

is deemed useful ($\Delta_q > \epsilon_1$), we add the question to the training dataset $a_{chosen} = a^q$. Otherwise we add the teacher action to the training dataset, as long as the teacher action is not significantly out of distribution for the student. To determine this, we measure the difference in probabilities of $a^s$ and $a^t$ under the student distribution: $\Delta_t = P_{\pi^s}(a^s|x) - P_{\pi^s}(a^t|x)$, and if $\Delta_t < \epsilon_2$, we choose the teacher action $a_{chosen} = a^t$. If neither conditions apply, we conclude that some information was missing but the question did not elicit it effectively and skip that datapoint. Additionally, if greedy decoding causes the teacher action to be more probable under the student distribution, i.e. $\Delta_t < 0$, we choose the teacher action $a_{chosen} = a^t$. The original student action $a_s$ becomes the rejected action $a_{rejected} = a^s$ in all cases.

**Direct Preference Optimization (DPO) for policy optimization:** To train the student model using the corrective data generated from the reflection process, we employ Direct Preference Optimization (Rafailov et al., 2024), a variant of reinforcement learning from human feedback (RLHF). DPO optimizes the policy using a closed-form objective that assumes a Bradley-Terry model (Bradley & Terry, 1952) of the user's reward function, enabling more stable training compared to traditional RLHF methods, which involve training a reward model and policy separately (Rafailov et al., 2024). We leverage DPO training over direct supervised finetuning, because we empirically find that it results in better performance (Section 6.3). We attribute this improvement to the KL-divergence constraint inherent in DPO, which prevents the model from deviating significantly from the base policy. This constraint is particularly beneficial in our context, as it helps avoid model forgetting when training on a relatively small dataset. For further regularization, we shorten the prompt to only contain a summary of rollout history, instead of the entire history. Each element of our dataset $[(x, a_{chosen}, a_{rejected}), ...]$ includes a chosen action $a_{chosen}$, a rejected action $a_{rejected}$, and a prompt $x$, which comprises the goal $g$, and a history of student actions $a^s_{0:k-1}$ and corresponding observations $o_{0:k-1}$ and any user feedback $u_{0:k-1}$. This ensures

that the student model is effectively aligned with user preferences, as demonstrated by the teacher, and asks questions when needed, while maintaining its foundational capabilities. In our experiments, the training data for Reflection-DPO contains about 37% questions, and 48% teacher actions. The rest of the datapoints deem the student action as appropriate.

# 6 Experiments

In this section, we analyze the performance of Reflection-DPO, state-of-the-art baselines, and ablations of Reflection-DPO. Our experiments highlight the importance of active questioning and its role in improving preference satisfaction in ADAPT. We also analyze ablations of Reflection-DPO, where we remove the reflection step, and replace DPO with supervised fine-tuning. We benchmark Reflection-DPO against the following baselines:

**ReAct** (Yao et al., 2023): A chain-of-thought style prompting by interleaving action prediction with thought prediction, a popular approach for using LLMs in planning.

**STaR-GATE** (Andukuri et al., 2024): Finetunes the student model using the top one-third of student rollouts that fulfill the highest fraction of user preferences. This reflects a common method of leveraging rollout rewards for finetuning LLMs.

**Baseline LLM**: Untrained zero-shot LLM, serves as a baseline to measure the benefit of training and question-asking capabilities.

**'Always Ask' LLM**: Variant of baseline LLM that asks a question before *every* action.

**'Never Ask' LLM**: Variant of the baseline LLM that *never* asks a question, serving as a lower bound of performance without any preference information on ADAPT tasks.

**Teacher LLM**: LLM with access to privileged user preferences, providing an upper bound on expected performance when learning from it. Does not need to ask any questions.

Both Qwen-2.5-72B and Llama 3.1-70b-Instruct show comparable performance on ReAct, our strongest baseline, as shown in Table 1. We choose Llama 3.1-70b-Instruct for all our experiments. We add decoder constraints, in the form of context-free grammar (Geng et al., 2023), to limit outputs to valid scene nodes and actions to all baselines in ADAPT tasks. This minimizes hallucinations and improves performance across the board for all approaches. While ADAPT allows open-set action verbs, we limit them to 82 plausible meal-related actions, like dice and peel, while keeping arguments and results open-set. Despite these constraints, nearly 60k valid actions remain available at any timestep. We evaluate generalization to unseen environments over both seen and unseen personas. We cross-validate across the 16 personas available in ADAPT, using 12 personas for training and 4 for evaluating generalization to unseen persona. The training mechanisms for Reflection-DPO, its ablations, and STaR-GATE, use a total of 192 trajectories, composed of 16 interactions with each of the 12 personas. This results in 1500 training datapoints for STaR-GATE, and about 5000 datapoints for Reflection-DPOand its variants. Note that STaR-GATE has about 1/3 of the datapoints by design (top 1/3 trajectories). We use Low Rank Optimization (Hu et al., 2022) with a rank of 4 for all models for efficient and regularized training. We set the reflection hyperparameters to $\epsilon_1 = 0.2$ and $\epsilon_1 = 0.5$ to preserve informative questions without allowing them to dominate over physical actions. This configuration yields a dataset composed of 37% questions and 48% physical actions.

## 6.1 Quantitative Analysis

**Metrics:** We measure performance using two metrics: preference satisfaction rate and the number of questions asked. In the ADAPT tasks, we evaluate a trajectory by the preferences satisfied $p_+$ and violated $p_-$, resulting in a preference satisfaction rate $\frac{p_+}{p_+ + p_-}$. A preference is considered satisfied only if the relevant subtask is completed, such as ensuring a tea bag is added to water to make tea. Subtask failure, such as not making tea when instructed to "make tea and toast for breakfast," results in all related preferences being violated. Thus the preference satisfaction rate reflects both task completion and percentage of preferences satisfied. Additionally, we measure the number of questions each approach asks.

| Method | Preference Satisfaction Rate ↑ | | Num. Questions Asked | |
|---|---|---|---|---|
| | Seen Persona | Unseen Persona | Seen Persona | Unseen Persona |
| 'Never Ask' LLM | 27.5%±0.7% | 28.6%±1.2% | 0.0±0.0 | 0.0±0.0 |
| Baseline LLM | 34.7%±0.8% | 34.3%±1.3% | 2.3±0.0 | 2.3±0.1 |
| ReAct | 36.1%±0.8% | 36.8%±1.4% | 2.3±0.0 | 2.4±0.1 |
| ReAct (Qwen) | 36.4%±1.5% | 39.5%±2.6% | 2.6±0.1 | 2.7±0.2 |
| STaR-GATE | 34.1%±0.8% | 33.5%±1.4% | 2.0±0.0 | 2.0±0.0 |
| Reflection-DPO | **44.1%**±1.0% | **42.9%**±1.6% | 9.8±0.1 | 9.5±0.2 |
| 'Always Ask' LLM | 52.1%±0.9% | 50.8%±1.6% | 22.2±0.1 | 22.4±0.2 |
| Teacher LLM | 65.5%±1.1% | 65.5%±1.9% | 0.0±0.0 | 0.0±0.0 |

Table 1: Comparison of Reflection-DPO and baselines for average number of questions asked and preference satisfaction rate.

**Reflection-DPO is able to elicit user preferences even for unseen personas.** Table 1 shows a comparison of Reflection-DPO against all baselines, reported through the mean and standard error for both seen and unseen persona. Reflection-DPO achieves a preference satisfaction rate of 44.1% for seen personas and 42.9% for unseen personas, outperforming all baselines, including ReAct (36.1% seen, 36.8% unseen) and STaR-GATE (34.1% seen, 33.5% unseen). This indicates that Reflection-DPO effectively generalizes without overfitting to the training personas. This shows that Reflection-DPO is able to elicit and adapt to user preferences better than leading zero-shot baseline ReAct, as well as STaR-GATE which is finetuned on successful past trajectories. Note that the performance is similar between seen and unseen persona because the training data contains no information about the specific persona it is being trained on. As a result, Reflection-DPO is able to learn how to elicit and adapt to preferences for any user, seen or unseen equally well.

**Asking questions is critical to high-performance on ADAPT.** The 'Always Ask' model shows that asking questions significantly improves performance, achieving a satisfaction rate of 52.1% for seen personas and 50.8% for unseen personas. However, it asks an average of 22 questions per interaction, which is impractical and overwhelming in real-world scenarios. In contrast, Reflection-DPO asks about 10 questions per interaction, efficiently eliciting preferences and approaching the performance of 'Always Ask' with fewer questions. Baseline models, such as the Baseline LLM and ReAct, ask only about 2-3 questions, which is insufficient for uncovering preferences and 'Never Ask' performs the worst. This indicates that ADAPT requires active questioning, and Reflection-DPO is able to elicit preferences.

**Reflection-DPO learns from an imperfect Teacher.** Even the privileged Teacher LLM, with a satisfaction rate of 65.5% for both seen and unseen personas, does not achieve perfect performance due to the long-horizon nature of tasks. For example, we observe that the Teacher might miss some steps, resulting in failures, such as retrieve butter and toast the bread but forget to butter the toast. Additionally, the freedom given to the planner to name entities resulting from freeform actions often mislead it into believing that a step is taken. Baseline-LLM without any preferences performs significantly worse (34.7% seen, 34.3% unseen) compared to Teacher, but Reflection-DPO is able to bridge this gap (44.1% seen, 42.9% unseen). This shows that even with an imperfect Teacher, Reflection-DPO is able to learn to follow and elicit preferences in complex, long-horizon ADAPT tasks. There may be further improvements through reinforcement learning and learning from experience, bridging the gap to the Teacher and even outperforming it, which we leave for future work.

### 6.2 Qualitative Analysis of Questions asked

**Reflection-DPO adapts questioning to preferences.** Reflection-DPO learns to ask more questions when more preferences are applicable, probing further when users have more task-related preferences. In contrast, baselines ask a similar number of questions regardless of the interaction context (Figure 3).

On average, each task has 5-6 relevant preferences, and Reflection-DPO asks 4-5 extra questions, while baselines fall short by 2-3 questions. Moreover, we observe that while Reflection-DPO asks extra questions (examples discussed in Appendix A), they are reasonable preferences that other personas could have had. However we observe that Reflection-DPO asks

| Method | Preference Satisfaction Rate ↑ | | Num. Questions Asked | |
| --- | --- | --- | --- | --- |
| | Seen Persona | Unseen Persona | Seen Persona | Unseen Persona |
| Reflection-DPO | **44.1%**±1.0% | **42.9%**±1.6% | 9.8±0.1 | 9.5±0.2 |
| Ablation: no Reflection | 17.5%±0.7% | 16.5%±1.2% | 1.0±0.0 | 0.9±0.0 |
| Ablation: no DPO | 35.3%±0.8% | 34.9%±1.4% | 2.9±0.1 | 2.9±0.1 |

Table 2: Comparison of Reflection-DPO against ablated versions without the reflection mechanism or DPO training, and against a student model that asks a question before each action.

more questions than strictly needed, which usually helps adapt to user preferences that the agent has no prior knowledge about. We leave it to future work to use knowledge from past interactions to further reduce questions.

**Reflection-DPO asks informative questions.** Qualitatively, we observe that Reflection-DPO asks higher-quality, and more informative questions than baselines. For example, at start of an interaction, it asks open-ended questions, such as "Do you want any toppings on your cereal?" and provides examples to clarify, such as "Would you like your coffee with any creamers, sweeteners, or flavorings, such as sugar, milk, or vanilla syrup?". When fewer options exist, it sim-

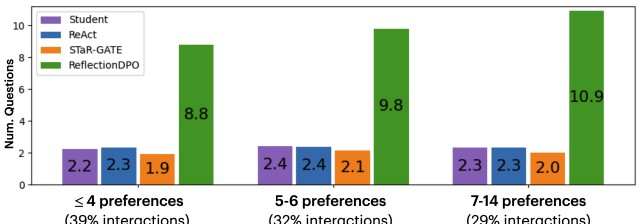

Figure 3: Num. questions asked over interactions with varying number of preferences show that Reflection-DPO asks more questions when more preferences exist.

plifies questions to binary responses, such as "Do you want milk in your coffee?". In contrast, baselines struggle to ask sufficient and prioritized questions. They can ask disambiguation questions when multiple options are present, but often fail to elicit all relevant user preferences. The 'Always Ask' LLM covers more preferences but gets sidetracked by questions about the world state or object availability, such as "Is the stove set to medium heat?". This information does not need user-feedback and is already available in the environment, resulting in missed opportunities to elicit actual preferences, which prevent the model from closing the gap to the Teacher. We include a discussion of *when* during a trajectory different methods adapt to preferences in Appendix A.

## 6.3 Ablations

Reflection-DPO differs from classical Dagger in two ways: the use of a reflection mechanism and DPO-based training. Table 2 studies the impact of both of these components.

**Reflection mechanism enhances questioning.** We compare Reflection-DPO to a "no Reflection" version that uses teacher actions directly for training. Without reflection, the preference satisfaction rate drops to 17.5% for seen personas and 16.5% for unseen personas, compared to 44.1% and 42.9% with reflection. The teacher data contains no questions, hence "no Reflection" asks few questions, failing to uncover preferences.

**DPO training improves generalization with a small training dataset.** We also evaluate a "no DPO" version of Reflection-DPO, trained with supervised finetuning on the chosen actions. This version achieves a satisfaction rate of 35.3% for seen personas and 34.9% for unseen personas, lower than Reflection-DPO. The large action space and long-horizon tasks in ADAPT create a vast input prompt space, leading to out-of-distribution evaluations, even for seen personas and tasks. The "no DPO" variant overfits to the small training dataset, despite extensive hyperparameter tuning. In contrast, DPO's KL-constraint regularizes training, preventing model forgetting and enabling better generalization.

## 7 Summary and Limitations

We present ADAPT, a benchmark for evaluating assistive agents at adapting long-horizon tasks to hidden user preferences, requiring preference elicitation by actively asking questions. We also propose a novel approach Reflection-DPO that finetunes an LLM for active questioning using a privileged teacher to identify preference-adhering actions, and a reflection step to identify when to ask questions. Reflection-DPO demonstrates strong performance in our experiments, eliciting user preferences for both seen and unseen personas and achieving higher satisfaction rates than baselines like chain-of-thought and supervised finetuning. This success is due to its ability to ask informative questions, efficiently uncovering user preferences with relatively few questions. Additionally, our ablation experiments demonstrate that the reflection mechanism significantly improves the LLM's questioning ability, and DPO training improves generalization, even with a small training dataset.

Ultimately, all question-asking models, including Reflection-DPO fall short of a privileged teacher's performance, indicating room for improvement on ADAPT. Most importantly, even the privileged teacher makes mistakes, in both preference satisfaction and task completion, indicating the need for learning stronger demonstrators, or learning from experience. Additionally, Reflection-DPO does not penalize questions, and as a result asks more questions than strictly needed. Future work can optimize the number of questions by introducing penalties and learning from prior experience with the same user.

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

# Appendix

## A  Preference elicitation performance over time

In this section, we examine *when* during an interaction each model adapts to various preferences. Figure 4 shows the average performance of each model throughout the course of an interaction, ultimately achieving the preference satisfaction rates reported in Section 6 at 100% completion. Our evaluation function is meant to be used for a complete trajectory, because preferences such as not adding ingredients and action ordering can wrongly appear to be satisfied when looking at a partial trajectory. To avoid such misleading effects, we measure the preference satisfaction rate at each step of an interaction by evaluating specifically those preferences which were satisfied at the end of the interaction. Doing so effectively captures the fraction of final performance achieved at all times over the span of a trajectory.

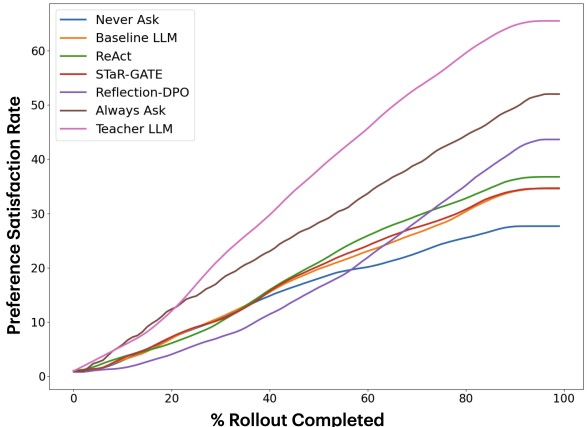

Figure 4: Comparison of preference satisfaction rate of different models over the interaction, showcasing when they adapt to differnt preferences.

Reflection DPO shows a slower improvement at the beginning, but continuously improves thereafter, surpassing all baselines. A subjective inspection shows that Reflection-DPO prioritizes exploration actions at the beginning, causing a delay in asking about and adapting to preferences. Later in the interaction, it keeps improving as a result of asking more questions. Reflection-DPO continues to probe for more fine-grained preferences, such as asking about which kind of nuts to use, once it knows that the user prefers to top their cereal with nuts. In contrast, all baselines show a steady improvement through the initial part of the interaction, but their improvement tapers off at the end.

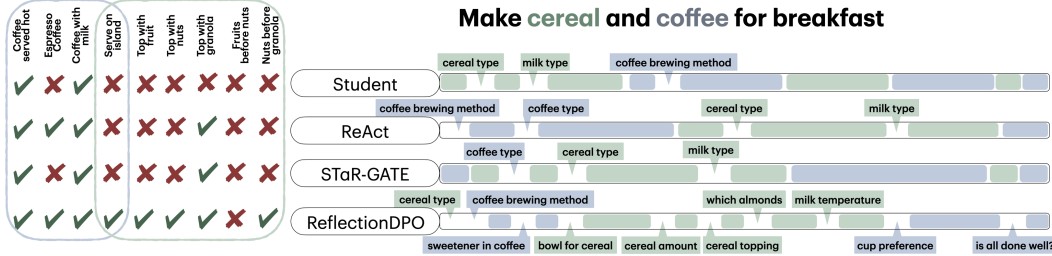

Figure 5: Example of actions taken through the course of a cereal and coffee task, with task component that each action is related to indicated by colors, and what different methods asked about. All baselines ask a similar set of 3-4 questions, but Reflection-DPO is able to probe further for more preferences.

Figure 5 shows an example comparing trajectories for all methods on a task of making cereal and coffee. Reflection-DPO is able to elicit a wide range of preferences by asking more general question at the beginning, followed by fine-grained follow-up questions towards

the end. We observe that the baselines consistently ask generic questions at the beginning, such as what kind of coffee the user would like, or they would like their eggs, but later on in the interaction their questions are more sparse, and are mostly related to disambiguating between instances of objects uncovered during exploration, such as various types of milk. Finally, the 'Always Ask' and Teacher LLM models show a steady improvement till the end, as they steadily adapt to preferences known a priori or elicited continuously throughout the course of the interaction.

## B    Satisfaction Rate per Question

Measuring satisfaction rate per question is challenging due to the lack of a "ground truth" for whether a question directly relates to a preference or satisfies multiple preferences. For instance, if the agent asks, "Do you like low-fat or whole milk?" and the user responds, "I am trying to be healthy, so low fat," the LLM might use this to offer healthier options like whole-wheat cereal in subsequent interactions. In this case, one question and response are responsible for multiple preference satisfaction.

To approximate question efficiency, we consider the ratio of preferences satisfied to user input, i.e. the number of questions and initial goal specification $\frac{p_+}{n_q+1}$. We report results comparing Reflection-DPO against baselines on this metric in Table 3. Note that this metric can favor models that satisfy preferences by chance without asking questions, or asking very few questions.

| Method | Seen Persona | Unseen Persona |
|---|---|---|
| Baseline LLM | 0.55 | 0.54 |
| ReAct | 0.57 | 0.56 |
| STaR-GATE | 0.59 | 0.59 |
| Reflection DPO | 0.26 | 0.25 |
| Always Ask | 0.12 | 0.12 |

Table 3: Satisfaction Rate per question of Reflection DPO compared against baselines

As expected, baselines that ask few questions like ReAct are more efficient at this metric, while Reflection-DPO and Always Ask are less efficient. Note that we currently don't explicitly penalize for asking extra questions, and merely aim to satisfy preferences (noted as a limitation in Section 7). In future work, we aim to increase the efficiency of our approach, aiming for models that can ask fewer questions while satisfying preferences.

## C    Details of ADAPT benchmark

The ADAPT benchmark includes 8 tasks and 16 personas in a simulation environment, where the agent can perform an open-set of actions. In this section we list out all the tasks, action templates and list of persona preferences in ADAPT.

### C.1    Actions

The full list of actions available to the planner, along with templates that the planner can use to invoke them, and the action-specific pre-conditions that must hold for successful execution of the action are enumerated in Table 4. Note that the existence of each entity referred in an action, excluding the result entities (those preceded by 'to get'), is a pre-condition for all actions.

| Action | Action Template | Conditions for success |
|---|---|---|
| ask | Ask ⟨ X ⟩ | |
| open | Open ⟨ X ⟩ | |
| close | Close ⟨ X ⟩ | |
| turn_on | Turn on ⟨ X ⟩ | X must be a cooking appliance |
| turn_off | Turn off ⟨ X ⟩ | X must be a cooking appliance |
| heat | Heat ⟨ X ⟩ | either X or the container containing X must be on a cooking appliance |
| search | Search ⟨ X ⟩ | |
| look_for | Look for ⟨ X ⟩ | |
| move_from | Move ⟨ X ⟩ from ⟨ Y ⟩ to ⟨ Z ⟩ | Y must contain X |
| move | Move ⟨ X ⟩ to ⟨ Y ⟩, Serve the object ⟨ X ⟩ at ⟨ Y ⟩, Place the object ⟨ X ⟩ at ⟨ Y ⟩ | |
| pour | Pour ⟨ X ⟩ from ⟨ Y ⟩ to ⟨ Z ⟩, Pour ⟨ X ⟩ from ⟨ Y ⟩ into ⟨ Z ⟩ | Y must contain X |
| mix | Mix all items in ⟨ X ⟩ to get ⟨ Y ⟩ | X must be a container |
| cook | Cook items in ⟨ X ⟩ to get ⟨ Y ⟩ | X must be on a cooking appliance |
| chop | Chop ⟨ X ⟩ to get ⟨ X ⟩ | X must be on a chopping surface |
| freeform | ⟨ X ⟩ items in ⟨ Y ⟩ to get ⟨ Z ⟩ | Y contain edible contents, and Z must be a container |
| freeform | ⟨ X ⟩ the object ⟨ Y ⟩ to get ⟨ Z ⟩ | Y must be edible, and Z must be a container |
| done | Declare Done | |

Table 4: All actions available to the planner, the associated templates expected from the planner, and action-specific pre-conditions.

## C.2 Tasks

The 8 tasks in ADAPT are:

1. Prepare eggs for breakfast
2. Prepare omelette for breakfast
3. Prepare cereal for breakfast
4. Make toast and coffee for breakfast
5. Make yoghurt parfait for breakfast
6. Make tea and eggs for breakfast
7. Make cereal and coffee for breakfast
8. Prepare tea and toast for breakfast

## C.3 Persona Preferences

ADAPT consists of a total of 16 personas. Each persona is composed of a set of 10-19 preferences. Following is the list of preferences for each persona:

- Persona 1:
  1. Prefer hearty scrambled eggs or omelettes fitting a high-protein high-calorie diet
  2. Add whole milk while preparing eggs
  3. Beverages such as coffee should be black, without milk or sugar
  4. Beverages such as tea should be black, without milk or sugar
  5. Beverages such as coffee should be served iced
  6. Beverages such as tea should be served iced
  7. Use full-fat milk or whipped cream to maintain a high-fat high-calorie diet

8. Top sweet breakfasts, such as french toast, cereal, pancakes, oatmeal, yoghurt parfait, with whipped cream to add some extra calories
9. Add cheese or avocado to savory breakfasts to add some extra calories
10. Stainless steel or cast iron cookware and durable utensils
11. Cream or avocado are great sides when making eggs
12. Dining room or patio dining locations

- Persona 2:

  1. Eggs must be loaded with toppings such as vegetables
  2. Eggs must be loaded with toppings such as cheese
  3. Pair a toast with spread
  4. Add nuts as a topping to sweet breakfasts, such as oatmeal, yoghurt parfait, pancakes, french toast, cereal
  5. Sprinkle cinnamon on sweet breakfasts, such as oatmeal, yoghurt parfait, pancakes, french toast, cereal
  6. Drizzle honey or maple syrup on sweet breakfasts, such as oatmeal, yoghurt parfait, pancakes, french toast, cereal
  7. Add fresh herbs to eggs
  8. Always use cheese and vegetables together, if making eggs
  9. When preparing cereal, add cereal first then milk, in that order
  10. Prefers to dine on the patio

- Persona 3:

  1. Vegan diet using egg substitutes
  2. Eggs should be prepared with a side of vegetables
  3. Avocados are a favorite as a side, and especially necessary on toast
  4. Beverages like coffee and tea are preferred is preferred black without any added sugars or creamers
  5. To maintain a vegan diet, use vegan alternatives to milk
  6. To maintain a vegan diet, use vegan alternatives to cheese when cooking eggs
  7. To maintain a vegan diet, use vegan alternatives to yoghurts
  8. A side of berries is a nice complement for any breakfast meal, no matter whether it is savory or sweet
  9. A side of nuts is a nice complement for any sweet breakfast
  10. Serve food directly at the standing desk to start the work day right away

- Persona 4:

  1. If eggs are eaten, make them into an omelette with at least 3 spices
  2. Add at least two vegetables when making eggs
  3. Prefer using butter, for example on toast
  4. Prefer using butter, for example when cooking eggs
  5. Prefer beverages like tea and coffee served hot
  6. Prefer beverages like tea and coffee served hot with milk and sugar
  7. Prefer a hint of cardamom in tea and coffee
  8. If yogurt is eaten, prefer it plain
  9. Add a hint of cardamom to sweet breakfasts, such as french toast, cereal, pancakes, oatmeal, yoghurt parfait
  10. Add nuts as a topping to sweet breakfasts, such as french toast, cereal, pancakes, oatmeal, yoghurt parfait
  11. Top eggs and toast with cheese
  12. Use chili as a garnish or provide hot sauce as a condiment
  13. Fresh herbs like cilantro or mint make a lovely garnish for eggs
  14. Serve breakfast at the formal dining room or outdoor patio

- Persona 5:

  1. Eggs should be accompanied by crispy bacon
  2. Eggs should be accompanied by toast
  3. Only use traditional full calorie sugars in sweet breakfasts, such as french toast, cereal, pancakes, oatmeal, yoghurt parfait

4. Only use full fat dairy milk
5. Oatmeal cooked with milk
6. Add a touch of sweetness to non-savory breakfasts such as pancakes, waffles, oatmeal, yoghurt parfait, etc.
7. Sweet breakfasts like french toast, cereal, yoghurt parfait, oatmeal, pancakes should be topped with maple syrup or honey
8. Sweet breakfasts french toast, cereal, yoghurt parfait, oatmeal, pancakes topped with nuts
9. Prefer flavored yoghurt
10. Yoghurt mixed with granola
11. Beverages such as tea and coffee preferred hot
12. Beverages such as tea and coffee preferred with milk and sugar
13. Melted cheese is a favorite addition to savory breakfast dishes, such as eggs
14. A sprinkle of chopped parsley or chives would be a nice touch to eggs
15. Serving beverages first would be a good approach, so it is ready to be had as soon as possible
16. Use butter to cook eggs and top toast
17. Prepare one of sausage, hash browns, or toast as a side when making eggs
18. Prefer to have a condiment as a side with savory breakfasts
19. Do not put vegetables in savory breakfasts

- Persona 6:
    1. Eggs must be scrambled, unless otherwise specified
    2. Incorporate fresh herbs for garnish on savory breakfasts
    3. Ron likes vegetables with eggs for added flavor
    4. Toast should be lightly buttered
    5. Toast should be topped with a spread of natural jam or honey
    6. Prefer lactose-free milk alternatives with cereal and oatmeal
    7. When it comes tofrench toast, cereal, yoghurt parfait, oatmeal, or pancakes, he has a sweet tooth and enjoys them topped with sweet ingredients like syrup, honey, or fresh fruit
    8. He likes sweet breakfasts, such as french toast, cereal, yoghurt parfait, oatmeal, pancakes topped with a sprinkle of cinnamon
    9. Yoghurt must be lactose free
    10. Yoghurt or oatmeal must be topped with fruit
    11. A sprinkle of fresh herbs is appreciated on savory breakfasts
    12. Add lactose-free cheese to eggs
    13. Add vegetables to eggs
    14. He prefers add milk first, then cereal rather than the other way around
    15. Sides like bacon or sausage, if available, are preferred with sweet breakfasts, like french toast, cereal, yoghurt parfait, oatmeal, pancakes
    16. He prefers his coffee to be served first to savor as he awaits his meal

- Persona 7:
    1. Poach eggs unless a different cooking method is specified
    2. Add milk when making scrambled eggs or omelette
    3. Sprinkle cheese on eggs
    4. Serve at least two condiments with eggs
    5. Top toast with cheese
    6. Use herbs on toast
    7. Use dairy milk when using any milk
    8. Add an extra splash of dairy milk when cooking sweet breakfasts, like cereal, yoghurt parfait, pancakes, french toast, oatmeal
    9. Use fruits as a side for sweet breakfasts, such as cereal, yoghurt parfait, pancakes, french toast, oatmeal
    10. Serve beverages like coffee and tea iced, not hot
    11. Use pour over coffee method
    12. Beverages served black, without any added sugars or creamers

    13. Use stainless steel or cast iron cookware

- Persona 8:
    1. Add only salt and pepper to eggs aside from oil
    2. Prefer healthy fats, such as olive oil to cook with
    3. Sweet breakfasts, such as pancake, waffles, french toast, oatmeal or yoghurt parfait, and even cereals, topped with fresh fruits instead of syrup or sugar
    4. Sweet breakfasts, such as pancake, waffles, french toast, oatmeal or yoghurt parfait, and even cereals, topped with nuts
    5. Sweet breakfasts, such as pancake, waffles, french toast, oatmeal or yoghurt parfait, and even cereals, topped with granola
    6. Always add fruits before nuts or granola
    7. Always add nuts before granola
    8. Always use dairy-based milks alternatives, and avoid plant-based milks
    9. Always use dairy-based yoghurts alternatives, and avoid plant-based yoghurts
    10. Beverages like tea and coffee are preferred hot, not iceds
    11. Use the espresso machine, if available, else pour over to make coffee
    12. Serve beverages such as tea and coffee with milk next to it, but not mixed in
    13. Have breakfast on the kitchen island

- Persona 9:
    1. Eggs scrambled or made into an omelette
    2. Eggs with vegetables like bell peppers, onions, and mushrooms
    3. Prefer side of fruits or avocado with eggs
    4. Toast toasted with spreads like avocado or hummus; instead of nut-based spreads
    5. add butter on toast
    6. Prefer herbal teas like peppermint or chamomile
    7. Beverages such as tea or coffee should be black, without milk or sugar
    8. Avoid nuts and nut based spreads on toast
    9. When adding milk to cereal, avoid nut-based milk, use a dairy-based alternative instead
    10. When using yoghurt, avoid nut-based yoghurt
    11. Add fruits as a topping to sweet breakfasts, such as oatmeal, yoghurt parfait, pancakes, french toast, cereal
    12. Serve items in a certain order, starting with the food first, then beverages
    13. Eat breakfast at a standing desk or kitchen island

- Persona 10:
    1. When cooking eggs, prefer them scrambled or made into an omelette
    2. Prefer a sprinkle of salt on eggs
    3. Prefer a sprinkle of pepper on eggs
    4. Prefer a topping of cheese on eggs
    5. Prefer a side of avocado with eggs
    6. Prefer toast made by toasting bread in a toaster
    7. Prefer bread with savory toppings like avocado, cheese or hummus
    8. Prefer oatmeal cooked a combination of water and a non-dairy milk alternative
    9. Prefer even sweet breakfasts, such as yoghurt parfait, pancakes, oatmeal, cereal, french toast, to be topped with savory toppings like nuts, seeds, cheese or avocado
    10. Prefer a slight sprinkle of salt even on sweet breakfasts, such as yoghurt parfait, pancakes, oatmeal, cereal, french toast
    11. Beverages like tea and coffee are preferred hot, not iced
    12. Beverages like coffee and tea are preferred is preferred black without any added sugars or creamers
    13. Prefers teas and coffees with caffeine, instead of decaf options or herbal teas
    14. Prefers a squeeze of lemon in her tea
    15. Prefers dairy-free cheese when topping eggs and toast
    16. Prefers the use of stainless steel or cast iron cookware
    17. Prefers to dine on the patio
    18. When preparing cereal, add cereal first then milk, in that order

- Persona 11:
    1. Prefer eggs made into an omelette, unless otherwise specified
    2. Add in some veggies like bell peppers or spinach to eggs
    3. My toast is topped with nut butter or avocado spread
    4. Coffee should be strong and iced
    5. Top pancakes or French toast with fresh fruit or a drizzle of honey
    6. Yogurt must be topped with granola
    7. Incorporate foods like nuts, and seeds in sweet breakfasts, such as french toast, cereal, yoghurt parfait, oatmeal, pancakes
    8. Add melted cheese to eggs, so it's easier to digest and adds a rich, creamy flavor
    9. Serve beverages first before foods
    10. When mixing cereal and milk, I prefer to pour the cereal in first - it helps prevent spills
    11. A side of fresh fruit or nuts with savory breakfast is appreciated to satisfy my sweet tooth
    12. Serve breakfast at my desk or on the outdoor patio
- Persona 12:
    1. Prefer gluten-free cereals
    2. Prefer gluten-free alternatives, when using bread for savory toast or french toast
    3. Prefer lactose-free milk alternatives
    4. Prefer toast without any spreads
    5. Do not use butter on toast, when cooking eggs, etc. since it is not lactose-free
    6. Sweet breakfasts, such as yoghurt parfait, pancakes, oatmeal, cereal, french toast etc. are preferred without any sugar, syrups or spreads
    7. Prefer yogurt alternatives made from non-dairy sources to avoid lactose
    8. Prefer beverages such as coffee served hot
    9. Prefer beverages such as tea served hot
    10. Prefer beverages such as tea without milk and sugar
    11. Prefer beverages such as coffee without milk and sugar
    12. Prefer cereal served either with lactose-free milk or eaten dry
    13. Prefer lactose-free alternatives, and adding cheese to breakfast dishes like eggs
    14. Prefer beverages to be served first, followed by food
    15. Prefer at least two vegetables like bell peppers, onions, or mushrooms as sides with eggs
    16. Prefer to first pour milk then add cereal, rather than the other way around
    17. Prefer to eat breakfast in his dining area or kitchen
- Persona 13:
    1. Eggs should be scrambled or over easy unless otherwise specified
    2. Does not take caffeine, prefers decaf coffee
    3. Does not take caffeine, prefers non-caffeinated teas
    4. Prefer butter, and nothing else, on toast
    5. Toast is a preferred side with any savory breakfast
    6. No jams on toast
    7. No spreads on toast except butter
    8. Add just a drizzle of honey, and nothing else, to sweet breakfasts, such as yoghurt parfait, pancakes, oatmeal, cereal, french toast
    9. No toppings (fruits, nuts, etc.) on sweet breakfasts such as yoghurt parfait, pancakes, oatmeal, cereal, french toast
    10. Beverages served hot, not iced
    11. Beverages served with a hint of honey
    12. Vegetables in omelettes or scrambled eggs
    13. Simple dishes rather than fancy ones
    14. Food served first, followed by beverages
    15. Breakfast served in the backyard or on the patio
- Persona 14:

1. Toast with sweet spreads like jams, marmalades, and nut butters
2. Add honey or maple syrup to sweet breakfasts, like oatmeal, yoghurt parfait, pancakes, french toast, cereal
3. Add fresh fruit to sweet breakfasts, like french toast, cereal, yoghurt parfait, oatmeal, pancakes
4. Add nuts to sweet breakfasts, like french toast, cereal, yoghurt parfait, oatmeal, pancakes
5. Add granola to yoghurt if available
6. Savory items served with a side of sautéed vegetables or hash browns
7. Beverages like tea and coffee served hot
8. Beverages like tea and coffee served with honey, when available or else a bit of sugar
9. Tea served with lemon
10. Melted cheese is a staple in her diet, especially when paired with eggs and veggies
11. Add at least two spices to savory breakfast dishes
12. Add herbs to savory breakfast dishes
13. Add a side of sauteed vegetables to savory breakfast
14. Use cast-iron cookware for breakfast dishes
15. Use expensive glassware for breakfast
16. Prefers a small dessert of fresh fruit and whipped cream with savory breakfasts

- Persona 15:

  1. Use whole-grain bread when making toast
  2. Make toast without butter or spreads
  3. Make coffee black, without milk or sweetner
  4. Sugar-free pancakes, waffles or French toast
  5. Add fresh fruits as toppings on sweet breakfasts, such as pancake, waffles, french toast, oatmeal or yoghurt parfait
  6. Prefer Plain yogurt over sweetened or flavored
  7. Prefer berries as the fruit topping with yoghurt or cereal
  8. Prefer eggs with a side of vegetables
  9. Add salt and pepper to eggs and vegetables
  10. Prefer eggs and vegetables seasoned with no other spices except salt and pepper
  11. Use non-stick pans when cooking
  12. Use silicone spatulas when cooking

- Persona 16:

  1. When cooking savory ingredients, such as eggs or vegetables, use at least two spices
  2. Use fresh herbs if cooking savory dishes like eggs, meats and vegetables
  3. When making eggs, scramble them or make an omelet
  4. When cooking eggs serve with a side of toast
  5. When making toast use whole grain bread
  6. When making coffee, serve black or with plant-based milk, if available, else black
  7. Mix oatmeal either only with water or with water and plant-based milk
  8. Add honey or maple syrup to oatmeal or sweet breakfasts, such as oatmeal, yoghurt parfait, pancakes, french toast, cereal
  9. Add fresh fruits, nuts, or seeds as toppings on sweet breakfasts, such as oatmeal, yoghurt parfait, pancakes, french toast, cereal
  10. Add cinnamon or cardamom on sweet breakfasts, such as oatmeal, yoghurt parfait, pancakes, french toast, cereal
  11. When using yoghurt prefer plant-based yoghurts
  12. When cooking or sauteeing breakfast meats, eggs, vegetables, etc. prefer using olive oil
  13. Serve tea without milk and sweetners
  14. Plant-based alternatives are always preferred, so use plant-based milk with cereal
  15. Use vegan cheese when making eggs, plant-based alternatives are always preferred
  16. Serve beverages first, when making breakfast
  17. Add milk first, then cereal, in that order
  18. Finally, serve food at a dining location

# D  Model Prompts

## D.1  Planner Prompt

The following is an example of the prompt the planner receives to predict the next action.

---

**Example Planner Prompt**

```
Source: system
Cutting Knowledge Date: December 2023
Today Date: 26 Jul 2024

You are an expert at task planning, and know how to provide assistance in a manner that
Person1 wants for preparing and serving breakfasts such as making cereal, pancakes,
toast, waffles, french toast, coffee, tea, etc. You have the ability to take the
following actions:
- Open <X>: open an instance of an articulated furniture or object. e.g. Open cabinet
- Close <X>: close an instance of an articulated furniture or object. e.g. Close cabinet
- Heat <X>: heat a container, which is located on a heating appliance. e.g. Heat pan_0
- Turn on <X>: turn on an appliance. e.g. Turn on stove_0
- Turn off <X>: turn off an appliance. e.g. Turn off stove_0
- Search <X>: search a container in the house. e.g. Search counter_0
- Look for <X>: Look for an object in the whole house. Use this with a generic category
name of an object, and not an object instance or phrase. Be sure to look for objects
one-at-a-time. e.g. Search apple
- Move <X> to <Y>: move an object X from wherever it currently is to the furniture or
location Y. e.g. Place plate_0 on table_2
- Mix all items in <X> to get <Y>: mix items that exist in a container, typically to
create a new entity. e.g. Mix all items in bowl_0 to get cake_batter
- Cook items in <X> to get <Y>: cook something in a stove, oven or other such appliance
to create a cooked version of that entity. e.g. Cook items in pan_0 to get scrambled_eggs
- Chop <X> to get <Y>: chop items on a cutting board to create a chopped version of that
entity. e.g. Chop apple_0 to get chopped_apple
- Pour <X> from <Y> to <Z>: pour an entity X from one container Y to another container
Z. e.g. Pour milk from milk_carton_0 to mug_0
- Move <X> from <Y> to <Z>: move content X of container Y to another container Z. e.g.
Move apple from apple_bag_0 to bowl_1
- <X> items in <Y> to get <Z>: freeform action to change object state, such as whisk,
heat, blend, etc. e.g. whisk items in pan_0 to get custard, brew coffee_grounds to get
brewed_coffee etc.
- <X> the object <Y> to get <Z>: freeform action to change object state, such as chop,
peel, crack, wash, wipe, etc. e.g. chop the object tomato_1 to get
finely_chopped_tomato, chop the object onion_0 to get sliced_onion, crack the object
egg_4 to get cracked_egg, etc.
- Ask <X>: ask a freeform question to the user to decide between multiple options and
make sure you adhere to the user's preferences. e.g. Ask "Would you like the eggs
sunny-side-up or scrambled?"

,
```

---

**Example Planner Prompt**

```
- Declare Done: indicate that the task is complete. Make sure to use this exactly once
at the end of the task.

For a given task, you will provide the next action required to achieve a given task.
Before each step you will provide your reason or intention behind your action.
e.g.
Thought: I need to find eggs to prepare an omelet
Action: Find eggs.
Do NOT repeat your last action.
Note that you must do the task in a way that the user prefers. Think of different
variations, modifications, sides, etc. applicable to the given task, and do the task in
a way that you think the user would prefer. You might know some things about the user,
which you should extrapolate from when possible, and if you don't have any related
information, you can ask the user a question. Be sure to only ask about their
preferences. Think about whether multiple options are available for a particular
ingredient, and think creatively about toppings and sides that the person might prefer.
Do not ask for help in finding things; the user may not know what objects exist in the
environment and where they might be located. Ask general questions to learn about the
user's preferences which can prove useful in preparing future breakfasts in addition to
the one at hand.

You have no prior information about user.

You will be performing tasks in a house with the following layout:
- kitchen (kitchen)
  - stove_0 (stove)
  - oven_0 (oven)
  - microwave_0 (microwave)
  - coffee_machine_0 (a drip coffee machine)
  - espresso_machine_0 (an espresso machine)
  - kettle_0 (an electric kettle)
  - toaster_0 (a classic 2-bread toaster)
  - dishwasher_0 (dishwasher)
  - island_0 (kitchen island next to countertop with two seats)
  - island_chair_0 (bar height chair next to kitchen island)
  - island_chair_1 (bar height chair next to kitchen island)
  - counter_0 (kitchen countertop next to the stove)
  - counter_1 (kitchen countertop next to the sink)
  - cabinet_0 (left kitchen cabinet above the counter)
  - cabinet_1 (right kitchen cabinet above the counter)
  - cabinet_2 (tall kitchen cabinet)
  - cabinet_3 (kitchen cabinet below the counter)
  - sink_1 (standard kitchen sink, contains water)
  - fridge_0 (fridge)
  - drawer_0 (top kitchen drawer to left of the stove)
  - drawer_1 (top kitchen drawer to right of the stove)
  - spice_cabinet_0 (small cabinet for spice bottles and related items)
- dining_room (dining room)
  - table_0 (wooden dining table with four chairs)
  - chair_1 (wooden dining chair)
  - chair_2 (wooden dining chair)
  - chair_3 (wooden dining chair)
- bedroom (bedroom)
  - bed_0 (queen-sized bed with a wooden frame)
  - dresser_0 (wooden dresser with six drawers)
  - nightstand_0 (wooden nightstand with two drawers)
  - lamp_0 (bedside lamp with a white shade)
  - desk_0 (work desk)
- living_room (living room)
  - chair_0 (recliner with a black leather upholstery)
  - sofa_0 (three-seater sofa with a brown leather upholstery)
  - coffee_table_0 (wooden coffee table with a glass top)
  - tv_0 (flat-screen TV with a 55-inch screen)
- bathroom (bathroom)
  - toilet_0 (white porcelain toilet)
  - sink_0 (white porcelain sink with a chrome faucet, contains water)
  - mirror_0 (large mirror with a wooden frame)
  - shower_0 (glass-enclosed shower with a chrome showerhead)
- patio (outdoor patio area)
  - table_1 (small patio table)

What is the next action required to achieve the task: Prepare omelette for breakfast?

,
```

```
                              Example Planner Prompt

Source: assistant
Action: Ask "Person1, would you like any fillings in your omelette, such as vegetables,
cheese, or meats, and if so, what would be your top choices?"

Source: environment
Observation:

Source: user
I would like cheese in my omelette.

Source: assistant
Action: Ask "user, what type of cheese would you prefer in your omelette, such as
cheddar, mozzarella, feta, or something else?"

Source: environment
Observation:

Source: user
I would prefer mozzarella cheese in my omelette.

Source: assistant
Action: Look for eggs

Source: environment
Observation: Found egg_carton_0 (carton of 12 eggs) in/on fridge_0, egg_6 (brown egg)
in/on egg_carton_0, egg_5 (brown egg) in/on egg_carton_0, egg_3 (brown egg) in/on
egg_carton_0, egg_2 (brown egg) in/on egg_carton_0, egg_4 (brown egg) in/on egg_carton_0

Source: user
You have not made anything yet.
What is the next step to complete the task: Prepare omelette for breakfast?

Source: assistant
Action:"

,
```

## D.2 Reflection Prompt

The following is an example of the prompt the student receives to predict a candidate question in the reflection step.

```
                             Example Reflection Prompt

Source: system
You are an expert at task planning, and can guide a robot on how to provide assistance
in a manner that user wants to Make cereal and coffee for breakfast.
Reflect on this difference between the action the robot should have predicted, and
actually predicted. Was there some knowledge about user's preferences, which if the
robot knew about, it would have predicted the expected action?\nExample: The robot
cannot predict that it should use almonds when making chia pudding, when it doesn't know
that user wants their chia pudding topped with almonds. If so, it could ask user a
question to clarify their preference first, such as 'Question: What toppings do you want
on your sweet breakfasts, like chia pudding?' If no question needs to be asked to
predict the expected action, you can say 'Question: None'. Answer with a single question
and do not provide any additional information or explanation."

Source: assistant
Action: Look for cereal

,
```

---

**Example Reflection Prompt**

```
Source: environment
Observation: Found cereal_box_4 (a box of cocoa puffs, contains cocoa_puffs_cereal)
in/on cabinet_0, cereal_box_5 (a box of classic corn flakes, contains
corn_flakes_cereal) in/on cabinet_0, cereal_box_0 (carton of fruit loops, contains
fruit_loops_cereal) in/on cabinet_0

Source: user
You have not made anything yet.
What is the next step to complete the task: Make cereal and coffee for breakfast?

Source: assistant
Instead of Ask "What type of cereal would you prefer for breakfast: cocoa puffs, classic
corn flakes, or fruit loops?", the robot was expected to perform Turn on
coffee_machine_0, because if the robot knew user better, it would have thought that
'Since Ron prefers his coffee to be served first to savor as he awaits his meal, I
should start by making coffee. To make coffee, I need to use the coffee machine.'
What question could the robot have asked user so that it could predict Turn on
coffee_machine_0 as its next step?"

'
```

## D.3  Persona Prompt

The following is an example of the prompt the persona model receives to answer the planner′s questions.

---

**Example Persona Prompt**

```
Source: system

You are teaching a household assistive robot in performing various assistive tasks in a
manner user would like. The robot may not know user's preferences, so your job is to
guide the robot to perform the given task for user. Be sure to guide the robot to make
only those dishes that the task calls for, e.g. if the task is to make a waffle do not
ask the robot to make other things, such as coffee. Answer direct questions regarding
your preferences, and not the availability or location of objects. In the latter case,
encourage the robot to search and explore different locations. Even if the robot makes
an irreversible error, be sure to provide a correction so that the robot does not repeat
it's mistakes the next time.

Given the current state of the house and what you know about user and the task at hand,
you will respond to the robot's last question concisely, and in first person, as if you
are user.

Environment State:
- kitchen (kitchen)
  - stove_0 (stove)
  - oven_0 (oven)
  - microwave_0 (microwave)
  - coffee_machine_0 (a drip coffee machine)
  - espresso_machine_0 (an espresso machine)
  - kettle_0 (an electric kettle)
  - toaster_0 (a classic 2-bread toaster)
  - dishwasher_0 (dishwasher)
  - island_0 (kitchen island next to countertop with two seats)
  - island_chair_0 (bar height chair next to kitchen island)
  - island_chair_1 (bar height chair next to kitchen island)
  - counter_0 (kitchen countertop next to the stove)
    - oil_bottle_0 (bottle of olive oil, contains olive_oil)
    - salt_box_0 (box of iodized salt, contains salt)
    - knife_block_0 (woode knife block with space for 9 knives)
      - knife_3 (bread knife with a serrated edge)
  - counter_1 (kitchen countertop next to the sink)
    - bread_0 (a sliced loaf of white sandwich bread, contains white_bread_slice)
    - bread_1 (a loaf of sprouted whole grain bread, contains whole_grain_bread_slice)
    - bread_2 (a sliced loaf of sweet banana and walnut bread, contains
    banana_bread_slice)
    - banana_0 (a ripe banana)
    - apple_0 (a honeycrisp apple)
  - cabinet_0 (left kitchen cabinet above the counter)

'
```

**Example Persona Prompt**

```
    - cereal_box_0 (carton of fruit loops, contains fruit_loops_cereal)
    - cereal_box_1 (carton of whole wheat shreds, contains whole_wheat_shreds_cereal)
    - cereal_box_4 (a box of cocoa puffs, contains cocoa_puffs_cereal)
    - cereal_box_5 (a box of classic corn flakes, contains corn_flakes_cereal)
    - pancake_mix_0 (a bag of classic pancake mix, contains pancake_mix)
    - pancake_mix_1 (a bag of low-sugar artificially-sweetened pancake mix, contains
    low_calorie_pancake_mix)
    - waffle_mix_0 (a bag of classic waffle mix, contains waffle_mix)
    - oatmeal_0 (packet of overnight oats, contains overnight_oats)
    - oatmeal_1 (box of Quaker 1-minute instant oats, contains instant_oats)
    - oatmeal_2 (plain rolled oats, contains plain_rolled_oats)
    - sugar_0 (small bottle of white sugar, contains white_sugar)
    - sugar_1 (packet of brown cane sugar, contains brown_cane_sugar)
    - sugar_2 (small bottle of loose calorie-free stevia, contains stevia)
    - honey_0 (bottle of honey, contains honey)
    - tea_bags_0 (box of green tea bags, contains green_tea_bag)
    - tea_bags_1 (box of herbal peppermint tea bags, contains peppermint_tea_bag)
    - tea_bags_2 (box of chamomile tea, contains chamomile_tea_bag)
    - tea_bags_3 (box of earl grey tea bags, contains earl_grey_tea_bag)
    - tea_bags_4 (box of english breakfast tea bags, contains english_breakfast_tea_bag)
    - coffee_0 (bag of regular ground coffee, contains coffee_grounds)
    - coffee_1 (bag of decaf ground coffee, contains decaf_coffee_grounds)
    - coffee_2 (jar of instant coffee, contains instant_coffee_powder)
    - beans_0 (can of black beans, contains black_beans)
    - beans_1 (can of pinto beans, contains pinto_beans)
    - beans_2 (can of garbanzo beans, contains garbanzo_beans)
    - pasta_0 (box of classic penne pasta, contains penne_pasta)
    - pasta_1 (box of whole wheat spaghetti pasta, contains spaghetti_pasta)
    - pasta_2 (box of lentil pasta, contains lentil_pasta)
    - almonds_0 (pack of raw almonds, contains raw_almonds)
    - almonds_1 (pack of roasted and salted almonds, contains salted_almonds)
    - pistachios_0 (pack of roasted and salted pistachios, contains salted_pistachios)
    - pistachios_1 (pack of chilli lime coated shelled pistachios, contains
    salted_pistachios)
    - walnuts_0 (pack of raw walnuts, contains raw_walnuts)
    - pumpkin_seeds_0 (pack of pumpkin seeds, contains pumpkin_seeds)
    - pine_nuts_0 (pack of pine nuts, contains pine_nuts)
    - cranberries_0 (pack of dry cranberries, contains dry_cranberries)
    - figs_0 (pack of dry figs, contains dry_figs)
    - raisins_0 (pack of raisins, contains raisins)
    - pecans_0 (pack of raw pecans, contains raw_pecans)
  - cabinet_1 (right kitchen cabinet above the counter)
    - bowl_0 (microwaveable glass bowl)
    - bowl_2 (microwaveable glass bowl)
    - bowl_3 (microwaveable glass bowl)
    - bowl_4 (white ceramic salad bowl)
    - bowl_9 (fancy porcelain china bowl with pink flowers)
    - bowl_10 (small steel bowl)
    - bowl_11 (small steel bowl)
    - plate_0 (flat white ceramic plate)
    - plate_4 (deep white ceramic plate)
    - plate_5 (deep white ceramic plate)
    - plate_6 (deep white ceramic plate)
    - plate_8 (fancy porcelain china plate with pink flowers)
    - cup_0 (fancy porcelain teacup with saucer)
    - cup_1 (fancy porcelain teacup with saucer)
    - cup_3 (simple white ceramic cup)
    - mug_0 (plain white mug)
    - french_press_0 (a glass french press)
    - pour_over_coffee_maker_0 (a chemex pour over coffee maker)
  - cabinet_2 (tall kitchen cabinet)
    - glass_0 (tall glass for drinking water)
    - glass_2 (tall glass for drinking water)
    - glass_3 (tall glass for drinking water)
    - glass_4 (short fancy cocktail glass)
    - wine_glass_0 (standard wine glass)
    - wine_glass_1 (standard wine glass)
    - wine_glass_3 (standard wine glass)
    - wine_glass_4 (tall champagne flute)
    - wine_glass_7 (tall champagne flute)
  - cabinet_3 (kitchen cabinet below the counter)
    - pan_0 (classic non-stick pan)
    - pan_1 (metal pan)
    - pot_0 (small aluminium pot)
    - pot_1 (large aluminium pot)

 '
```

**Example Persona Prompt**

```
    - cutting_board_0 (plastic cutting board)
    - cutting_board_1 (wooden cutting board)
    - tray_0 (serving tray)
    - box_1 (plastic contianer for leftovers)
    - box_2 (small microwaveable glass contianer for leftovers)
    - box_3 (large microwaveable glass contianer for leftovers)
  - sink_1 (standard kitchen sink, contains water)
    - sponge_0 (sponge for cleaning)
    - dish_soap_0 (bottle of dish soap, contains dish_soap)
  - fridge_0 (fridge)
    - water_bottle_0 (bottle of still water, contains water)
    - ice_tray_0 (ice tray with ice cubes, contains ice_cube)
    - milk_0 (carton of whole dairy milk, contains whole_milk)
    - milk_3 (bottle of almond milk, contains almond_milk)
    - milk_4 (jug of skim dairy milk, contains skim_dairy_milk)
    - yoghurt_0 (pack of plain non-fat greek yoghurt, contains greek_yoghurt)
    - yoghurt_2 (cup of vegan cashew-based yoghurt, contains vegan_yoghurt)
    - egg_carton_0 (carton of 12 eggs)
      - egg_0 (brown egg)
      - egg_1 (brown egg)
      - egg_6 (brown egg)
      - egg_7 (brown egg)
    - liquid_egg_0 (a box of vegan liquid egg substitute, contains vegan_egg_substitute)
    - butter_0 (block of butter, contains butter)
    - butter_1 (box of vegan butter, contains vegan_butter)
    - cheese_0 (slices of american cheese, contains american_cheese_slice)
    - cheese_1 (block of part skim mozzarella cheese, contains mozzarella_cheese)
    - tomato_0 (roma tomato)
    - bell_pepper_1 (red bell pepper)
    - onion_0 (yellow onion)
    - potato_0 (sweet potato)
    - jalepeno_0 (spicy jalepeno peppers)
    - lettuce_0 (head of iceberg lettuce)
    - strawberries_box_0 (a pint-sized box of organic strawberries, contains strawberry)
    - oranges_bag_0 (a bag of navel oranges, contains navel_orange)
    - avocado_0 (a ripe avocado)
    - mixed_berries_0 (a bag of frozen mixed berries, contains frozen_berry)
    - hash_browns_0 (a bag of ready-to-cook frozen hash browns, contains hash_brown)
    - hummus_0 (box of hummus, contains hummus)
    - salsa_0 (jar of salsa, contains salsa)
    - cheese_dip_0 (jar of cheese dip, contains cheese_dip)
    - mayonnaise_0 (jar of mayonnaise, contains mayonnaise)
    - jam_1 (jar of strawberry jam, contains strawberry_jam)
    - spread_0 (jar of orange marmalade, contains orange_marmalade)
    - spread_2 (jar of peanut butter spread, contains peanut_butter_spread)
    - hot_sauce_0 (bottle of habanero hot sauce, contains habanero_hot_sauce)
    - hot_sauce_1 (bottle of tabasco hot sauce, contains tabasco_hot_sauce)
    - bacon_0 (pack of breakfast bacon, contains bacon_strip)
    - parsley (box of fresh parsley, contains fresh_parsley)
    - chives (box of fresh chives, contains fresh_chives)
    - basil (box of fresh basil, contains fresh_basil)
    - oregano (box of fresh oregano, contains fresh_oregano)
    - dill (box of fresh dill, contains fresh_dill)
    - cilantro (a bunch of fresh cilantro, contains fresh_cilantro)
    - maple_syrup_1 (small bottle of artificially sweetened low-calorie maple syrup,
    contains low_calorie_maple_syrup)
  - drawer_0 (top kitchen drawer to left of the stove)
    - knife_0 (simple metal butter knife)
    - knife_1 (simple metal butter knife)
    - spoon_0 (simple metal dinner spoon)
    - spoon_2 (simple metal dinner spoon)
    - fork_2 (simple metal dinner fork)
    - fork_4 (simple metal dinner fork)
    - knife_7 (simple steak knife)
    - knife_8 (simple steak knife)
  - drawer_1 (top kitchen drawer to right of the stove)
    - ladle_0 (ladle to serve food)
    - serving_scoop_0 (scoop to serve food)
    - spatula_2 (plastic spatula)
    - spatula_3 (wooden spatula)
  - spice_cabinet_0 (small cabinet for spice bottles and related items)
    - pepper_0 (small bottle of pepper, contains pepper)
    - mixed_herbs_0 (small bottle of mixed herbs, contains mixed_herbs)

  '
```

**Example Persona Prompt**

```
    - garlic_powder_0 (small bottle of garlic powder, contains garlic_powder)
    - onion_powder_0 (small bottle of onion powder, contains onion_powder)
    - cinnamon_0 (small bottle of cinnamon, contains cinnamon)
    - nutmeg_0 (small bottle of nutmeg, contains nutmeg)
    - clove_0 (small bottle of clove, contains clove)
- dining_room (dining room)
  - table_0 (wooden dining table with four chairs)
    - napkins_0 (set of four cloth napkins)
    - salt_shaker_0 (table salt shaker, contains salt)
    - pepper_shaker_0 (pepper shaker, contains pepper)
  - chair_1 (wooden dining chair)
  - chair_2 (wooden dining chair)
  - chair_3 (wooden dining chair)
- bedroom (bedroom)
  - bed_0 (queen-sized bed with a wooden frame)
  - dresser_0 (wooden dresser with six drawers)
  - nightstand_0 (wooden nightstand with two drawers)
  - lamp_0 (bedside lamp with a white shade)
  - desk_0 (work desk)
- living_room (living room)
  - chair_0 (recliner with a black leather upholstery)
  - sofa_0 (three-seater sofa with a brown leather upholstery)
  - coffee_table_0 (wooden coffee table with a glass top)
  - tv_0 (flat-screen TV with a 55-inch screen)
- bathroom (bathroom)
  - toilet_0 (white porcelain toilet)
  - sink_0 (white porcelain sink with a chrome faucet, contains water)
  - mirror_0 (large mirror with a wooden frame)
  - shower_0 (glass-enclosed shower with a chrome showerhead)
- patio (outdoor patio area)
  - table_1 (small patio table)

Task: Make cereal and coffee for breakfast

User has the following preferences.
When cooking savory ingredients, such as eggs or vegetables, use at least two spices
Use fresh herbs if cooking savory dishes like eggs, meats and vegetables
When making eggs, scramble them or make an omelet
When cooking eggs serve with a side of toast
When making toast use whole grain bread
When making coffee, serve black or with plant-based milk, if available, else black
Mix oatmeal either only with water or with water and plant-based milk
Add honey or maple syrup to oatmeal or sweet breakfasts, such as oatmeal, yoghurt
parfait, pancakes, french toast, cereal
Add fresh fruits, nuts, or seeds as toppings on sweet breakfasts, such as oatmeal,
yoghurt parfait, pancakes, french toast, cereal
Add cinnamon or cardamom on sweet breakfasts, such as oatmeal, yoghurt parfait,
pancakes, french toast, cereal
When using yoghurt prefer plant-based yoghurts
When cooking or sauteeing breakfast meats, eggs, vegetables, etc. prefer using olive oil
Serve tea without milk and sweetners
Plant-based alternatives are always preferred, so use plant-based milk with cereal
Use vegan cheese when making eggs, plant-based alternatives are always preferred
Serve beverages first, when making breakfast
Add milk first, then cereal, in that order
Finally, serve food at a dining location

Look at the following interaction and provide a short answer to the robot's last
question based on user's preferences. If user is flexible in their preference, make a
choice arbitrarily, but make sure to tell the robot that usually user is flexible, and
options which they would be okay with. Make sure to be consistent with your previous
feedback.

Source: robot
Action: Look for cereal

Source: environment
Observation: Found cereal_box_4 (a box of cocoa puffs, contains cocoa_puffs_cereal)
in/on cabinet_0, cereal_box_5 (a box of classic corn flakes, contains
corn_flakes_cereal) in/on cabinet_0, cereal_box_0 (carton of fruit loops, contains
fruit_loops_cereal) in/on cabinet_0

Source: robot
Ask "What type of cereal would you prefer for breakfast: cocoa puffs, classic corn
flakes, or fruit loops?"

'
```

