# OpenReview forum: "ADAPT: Actively Discovering and Adapting to Preferences for any Task"
_colmweb.org/COLM/2025/Conference — COLM 2025_

### Official Review · Reviewer_bZvY · 2025-05-12

**Rating:** 7
**Confidence:** 3
**Ethics Flag:** 1

**Summary:**

The paper is about adapting language models for any tasks by training it to actively ask preference questions. While this has been studied in previous research for different contexts (behavior adaptation, task oriented dialog, information seeking questioning), paper contributes (1) designed a benchmark to evaluate agent's capability to adhere to preferences (2) introduced reflection-DPO to produce "reflection" data  to fine-tune a student model to mimic a privileged teacher model. Proposed method is compared to the performance of existing methods (zero shot cot baseline, ReAct, star-gate (most related).

**Reasons To Accept:**

- Paper presents an interesting approach to adapting agents behavior. The choice of using preference and reflection-DPO (at steps) are interesting and novel to me.
     Strategies to obtain preference data are interesting.
- Experiment analysis and results shows the method works better (less questions asked, better preference satisfaction rate.
- Compared with various baselines (basic zero-shot cot, or adapted ones).
- Paper writing is clear and easy to understand.

**Reasons To Reject:**

na

---

> ### Author Response · Authors · 2025-06-02
>
> We thank the reviewer for their thoughtful and positive feedback, and the appreciation for the novelty of our approach, our evaluations and writing clarity.

---

> > ### Author Response · Authors · 2025-06-10
> >
> > Thank you for taking the time to review our submission. We noticed that the score for our paper changed from 9 to 7 during the rebuttal process, but we did not see any additional comments or concerns raised at that stage. We would really appreciate it if you could share any specific points that led to the adjustment. This would help us better understand your feedback and improve our work.

---

### Official Review · Reviewer_ojsc · 2025-05-14

**Rating:** 7
**Confidence:** 4
**Ethics Flag:** 1

**Summary:**

This paper proposes the ADAPT task, which involves adapting an agent’s action selection to user preferences by referencing dialogue history. It also introduces Reflection-DPO, which updates the agent’s behavior to reflect user preferences. The results show that the policy trained with Reflection-DPO can implicitly adapt to user preferences using only dialogue history.

**Questions To Authors:**

There are some areas where the related work and technical details could be expanded or clarified.

The design of the ADAPT task is interesting. However, the paper lacks an adequate review and contrast with prior work. For example, approaches that adapt dialogue policies based on the focus of user utterances have existed for quite some time. Additionally, frameworks have been proposed in the context of agent action selection in which appropriate actions are chosen based on dialogue history or surrounding context, especially in the absence of explicit user requests.

The ADAPT dataset is based on 16 manually created personas. It would be helpful to include a discussion of how these personas were selected. Presumably, they were designed to cover a diverse range of user types, but is there any rationale for selecting exactly 16 personas? Is there any justification that this number is sufficient?

I also have a question about the teacher model. Is the teacher model the same as the user simulator, or is it simply a language model with access to the gold persona? A naive implementation suggests it might be a dialogue model that includes the user simulator. Could the authors clarify this point?

Regarding the ε hyperparameter, were there any methods to avoid data leakage or heuristics when setting its value?

When evaluating the usefulness of questions, one could consider using action sampling results. Although this may increase training time, it could allow evaluation without the influence of the hyperparameters.

References

Yoshino, Koichiro, and Tatsuya Kawahara. "Conversational system for information navigation based on POMDP with user focus tracking." Computer Speech & Language 34.1 (2015): 275-291.

Tanaka, Shohei, et al. "Do as I Demand, Not as I Say: A Dataset for Developing a Reflective Life-Support Robot." IEEE Access 12 (2024): 11774-11784.

**Reasons To Accept:**

Overall, the paper was a unique and well-evaluated contribution.
The design of the ADAPT task is interesting.

**Reasons To Reject:**

There are some lacks of references.
There are some questions about data construction and implementation details.

---

> ### Author Response · Authors · 2025-06-02
>
> We thank the reviewer for the positive feedback on our work. We are pleased that you found the ADAPT task design to be unique and well-evaluated.  Below, we address specific questions and outline additional experiments conducted based on the suggestions provided.
>
> **Related work:**
>
> We have expanded the related work section to better contrast our approach with existing methods, as suggested by the reviewer. Thank you for highlighting these works.
>
> **Persona creation:**
>
> We have included all persona descriptions in Appendix B.3. The 16 personas were manually created and selected to represent a diverse range of user types, ensuring broad coverage of potential user preferences. While more personas can be added, we believe that the current set is diverse enough to enable a systematic study of adaptation to preferences.
>
> **Teacher model and user simulation:**
>
> Both the teacher model and the LLM-based user simulation are LLMs with access to ground truth user preferences, but they are prompted to perform different tasks. The teacher acts as an assistive agent, using preferences to plan tasks, while the user simulation responds to agent queries. The planner and persona prompts in Appendix C.1 and C.3 illustrate this distinction. We will further clarify this in the paper.
>
> **Reflection Hyperparameters:**
>
> The $ \epsilon $-hyperparameters in our reflection mechanism balance the questions demonstrated by reflection and the physical actions demonstrated by the teacher model. We tuned the hyperparameters on the train data (avoiding data leakage) to keep informative questions while not overshadowing the actions. We empirically tune our parameters ($ \epsilon_1 $=0.2 and $ \epsilon_2 $=0.5), resulting in 37% of the dataset being questions and 48% being physical actions. This analysis was done entirely on the train dataset, and hence ensured that there was no data leakage. In the final paper, we will add a result doing a hyperparameter sweep and cross-validation for thoroughness.
>
> **Alternative baseline without any hyperparameters:**
>
> We appreciate the idea of exploring variations of Reflection-DPO that do not use hyperparameters. We tried two variants of our reflection mechanism which would not require hyperparameters, and would instead directly compare sampled probabilities.
> First, we created a ‘Question Advantage’ baseline that predicts the question if it has a positive utility ($ P_{\pi_s} (a_t|q, x) - P_{\pi_s} (a_t|x) > 0  $).
> Second, we created a ‘Mean Probability’ version that treats the predictions as multiple future hypotheses (asking the question then predicting teacher action and directly predicting teacher action), and compares the average probability of each branch, choosing the question if probability of that branch ($ \frac{P_{\pi_s}(a_t|q, x) + P_{\pi_s}(q|x)}{2} $) is larger than the other ($ P_{\pi_s}(a_t|x) $). We evaluated both variations and empirically found their performance to be lower than our reflection mechanism.
>
>
> | Method        | Preference Satisfaction Rate || Num. Questions Asked ||
> |---------------|---------------|--------------|------------|-----------|
> |               |Seen Persona | Unseen Persona | Seen Persona | Unseen Persona |
> | Reflection DPO| 44.1% ± 1.0% | 42.9% ± 1.6% | 9.8 ± 0.1 | 9.5 ± 0.2 |
> | Question Advantage| 12.3% ± 1.2% | 9.6% ± 1.6% | 32.8 ± 0.6 | 33.7 ± 1.0 |
> | Mean Probability| 17.3% ± 1.3% | 19.3% ± 2.4% | 17.1 ± 0.5 | 18.2 ± 1.0 |
>
> We find that the ‘Question Advantage’ baseline asks far too many questions. It is unable to distinguish between questions that gain information versus questions that simply increase the probability of teacher action (by mentioning the relevant object, for example). This leads to over-emphasis on questions, with the majority of the dataset consisting of questions, resulting in a model that asked questions and didn’t take enough actions.
> The ‘Mean Probability’ version requires the tokenwise probabilities of an action generation and a question generation to be mutually comparable. This fails in practice due to the sensitivity of LLM probabilities to generation lengths and other extraneous syntactic variables. We posit that new breakthroughs in uncertainty calibration [1] can enable such hyperparameter-free methods to succeed, but we leave it to future work to explore efficient calibration mechanisms for this domain.
>
> We appreciate the reviewer's suggestion and will include these results in an appendix section on alternatives to our reflection mechanism.
>
> [1] Liu, Xiaoou, et al. "Uncertainty quantification and confidence calibration in large language models: A survey." arXiv preprint arXiv:2503.15850 (2025).

---

> > ### Comment · Area_Chair_RURA · 2025-06-06
> > **Acknowledging Author Response**
> >
> > Dear Reviewer ojsc,
> >
> > Please take a moment to acknowledge the author's response and ask any follow-up questions that you may have. E.g., reviewers have expanded the related work and added more details on how the personas were selected. Has this response answered these questions? If yes, then please acknowledge and update your review accordingly. If not, then please let the authors know this with your justification.
> >
> > Your response is especially important since you have a borderline acceptance recommendation.
> >
> > Your AC

---

> > ### Comment · Reviewer_ojsc · 2025-06-10
> > **Reply to authors**
> >
> > Thank you for addressing my questions.
> > The authors have responded appropriately to my concerns, and I would like to raise my score accordingly.
> > I also appreciate the additional experiments conducted in support of the proposal.
> > While I had not intended to request further experiments, I believe that comparing the proposed method with other naïve alternatives is a valuable addition.

---

### Official Review · Reviewer_tqzp · 2025-05-25

**Rating:** 7
**Confidence:** 4
**Ethics Flag:** 1

**Summary:**

## Empiricism, Data, and Evaluation
- Table 1 is informative - showing the different results with error bars with different methodologies
- Table 1 has satisfaction rate and num questions asked, it seems like a metric the paper would like to measure is also the increase in satisfaction rate per question, since the "Always Ask" LLM is considered "question inefficient". Analysis of the additional questions asked during Reflection-DPO indicate the model is learning which questions are more valuable to ask, and how to better formulate those questions.

## Technological Impact
- Including prompts, open sourcing code, and detailed descriptions of setup all help future researchers with experiment design by providing reference materials for reproduction.
- Developing a reliable and well curated benchmark for preference adherence in long-horizon tasks is valuable to the research community.

## Ambition, Vision, Forward-outlook Progress
- The paper provides a benchmark which increases the complexity of task measured by increasing the length and complexity of interaction.

## Understanding Depth, Principled Approach
- It would be nice to see performance results from a wide array of SoTA LLM as agents on the ADAPT benchmark.
- One criticism of the paper may be that the benchmark uses an LLM simulated user, but this seems to be acceptable and necessary for the proliferation and reproducibility of the benchmark. Real-user preferences were transcribed

## Clarity, Honesty, and Trust
- Applaud the authors for releasing code which can reproduce the results in the paper and run the benchmark.

**Reasons To Accept:**

- Publishing new benchmark with coverage over a more complex set of long horizon tasks is a welcome contribution.
- The paper adds to the library of work discussing how to systematically evaluate assistive agents.
- Reflection-DPO description and methodology are informative and novel. Analysis of the additional questions asked during Reflection-DPO indicate the model is learning which questions are more valuable to ask, and how to better formulate those questions.

**Reasons To Reject:**

- Authors may want to increase complexity or difficulty of the ADAPT benchmark.
- Authors may want to make the ADAPT benchmark more relevant by adding more tasks in varying domains.
- Authors may want to add more baseline SoTA model results on the ADAPT benchmark

---

> ### Author Response · Authors · 2025-06-02
>
> We thank the reviewer for their detailed review and the appreciation of the value of the ADAPT benchmark creation and open-source sharing. We are grateful for your positive remarks regarding our benchmark's contribution and the novel methodology of Reflection-DPO. Below we address some of the concerns raised:
>
> **Increasing Complexity and Adding Tasks:**
>
> Our primary goal in designing ADAPT tasks was to ensure sufficient task complexity with many possible preferences, while keeping tasks long-horizon and open-ended. Expanding to different domains and adding new tasks currently requires resources, including manual effort for creating reward functions and computational resources for scaling experiments. Although modern LLMs could assist, manual verification and correction are often necessary to ensure correctness. We plan to address this expansion in future work and will note this limitation in Section 7.
>
>
> **Performance Results with SoTA LLMs:**
>
> Our evaluations focused on how different approaches using the same LLM perform on ADAPT, including prior research on active task-oriented questioning (STaR-GATE) and strong planning baselines (ReAct). We chose Llama 3.1 as the base LLM due to its satisfactory performance.
>
> We agree that evaluating ADAPT across different LLM models would provide valuable insights. We are currently running experiments with the Qwen-2.5-72B model on ReAct, our strongest baseline, and will update this rebuttal with preliminary results. Final results will be included in the paper.
>
> **Measuring Satisfaction Rate per Question:**
>
> Measuring satisfaction rate per question is challenging due to the lack of a "ground truth" for whether a question directly relates to a preference or satisfies multiple preferences. For instance, if the agent asks, "Do you like low-fat or whole milk?" and the user responds, "I am trying to be healthy, so low fat," the LLM might use this to offer healthier options like whole-wheat cereal in subsequent interactions. In this case, one question and response are responsible for multiple preference satisfaction.
>
> To approximate question efficiency, we consider the ratio of preferences satisfied to user input, i.e. the number of questions and initial goal specification $ \frac{p_+}{n_q + 1} $ . However, this metric can favor models that satisfy preferences by chance without asking questions, or asking very few questions. With this caveat, here are the "efficiency of questions" per baseline:
>
>
> | Method       | Seen Persona | Unseen Persona |
> |--------------|--------------|----------------|
> | Baseline LLM | 0.55         |     0.54      |
> | ReAct        | 0.57         |     0.56      |
> | STaR-GATE    | 0.59          |     0.59        |
> | Reflection DPO| 0.26         |     0.25      |
> | Always Ask     |  0.12     |     0.12     |
>
> As expected, baselines that ask few questions like ReAct are more efficient at this metric, while Reflection-DPO and Always Ask are less efficient. Note that we currently don’t explicitly penalize for asking extra questions, and merely aim to satisfy preferences (noted as a limitation in Section 7). In future work, we aim to increase the efficiency of our approach, aiming for models that can ask fewer questions while satisfying preferences. We will add this metric and discussion to Appendix.
>
>
> **Use of LLM Simulated User:**
>
> We use an LLM, prompted with a set of preferences, to simulate a user to enable closed-loop experimentation and reproducibility on the benchmark. The simulated user enables us to test different baselines with open-ended questions, and judge their performance in a systematic manner.

---

> > ### Comment · Reviewer_tqzp · 2025-06-03
> > **Response to Official Comment by Authors**
> >
> > We thank the authors for addressing the concerns raised in the Official Review. We believe all the concerns have been addressed.

---

### Author Response · Authors · 2025-06-02

We thank the reviewers for their feedback, and appreciate that the reviewers find our approach novel (R1, R2, R3) and contributions well-evaluated (R2, R3), and appreciate the value of ADAPT benchmark and code release (R1).

In this rebuttal, we address concerns raised by the reviewers. Specifically:
1. We are running new experiments with the Qwen-2.5-72B model on ReAct to compare other SoTA LLMs at  ADAPT.
2. We add two additional hyperparameter-free variations of Reflection-DPO, and plan to add a hyperparameter sweep to the final paper to address concerns about data leakage and heuristics.
3. We add a new metric ‘Question Efficiency’ to study the efficiency of different approaches at satisfying preferences.

---

> ### Author Response · Authors · 2025-06-10
>
> Our preliminary evaluations on using Qwen2.5-72B as a backbone for ReACT show similar results as Llama3.1, with marginally better performance by asking a slightly larger number of questions. The results are as follows:
> | Method        | Preference Satisfaction Rate || Num. Questions Asked ||
> |---------------|---------------|--------------|------------|-----------|
> |               |Seen Persona | Unseen Persona | Seen Persona | Unseen Persona |
> | Llama3.1-70B | 36.1% ± 0.8% | 36.8% ± 1.4% | 2.3 ± 0.0 | 2.4 ± 0.1 |
> | Qwen2.5-72B | 36.4% ± 1.5% | 39.5% ± 2.6% | 2.6 ± 0.1 | 2.7 ± 0.2 |
> | ReflectionDPO (w/ Llama3.1-70B) | 44.1% ± 1.0% | 42.9% ± 1.6% | 9.8 ± 0.1 | 9.5 ± 0.2 |
>
>
> We thank the reviewers and the area chair for their time and thoughtful feedback. We hope that our responses and additional results have addressed the concerns raised, and we respectfully invite the reviewers to reconsider their evaluations in light of these clarifications.

---

### Comment · Area_Chair_RURA · 2025-06-06
**Discussion Period Ends on June 10th**

Dear Reviewers,

The discussion period ends on June 10th. If you have any other follow-up questions or concerns, then please post them soon so authors have a chance to respond. Please also look at other reviews and follow-up messages.

Thank you,

Your AC

---

### Decision · Program_Chairs · 2025-07-08

**Decision:**

Accept

**Comment:**

This paper proposes a benchmark ADAPT where a student LLM agent needs to jointly work with a teacher LLM agent, which has privileged information in the form of ground-truth preferences. The student agent must actively ask questions to learn the preferences of the teacher and act accordingly. The paper also proposes an algorithm, Reflection DPO, that uses preference data generated via students' active questioning and uses it to fine-tune the student. Results in Table 1 show that ReflectionDPO achieves good accuracy while not asking a lot of questions.

Overall, all reviewers found this paper to be good and recommended acceptance. Reviewers overall found the paper to be interesting. One potential issue can be that the baseline that always asks questions does 8% better than ReflectionDPO and may still be preferred when the number of questions asked is less important than quality. It would have been a much stronger result if ReflectionDPO could match its performance while keeping the number of questions lower.

However, I think this is an interesting benchmark and the field will benefit from these results, so I am still recommending acceptance.